# Nitrogen-Doped Carbons Derived from Imidazole-Based Cross-Linked Porous Organic Polymers

**DOI:** 10.3390/molecules26030668

**Published:** 2021-01-27

**Authors:** Wojciech Kiciński, Sławomir Dyjak

**Affiliations:** Institute of Chemistry, Military University of Technology, 2 Kaliskiego Str., PL 00-908 Warsaw, Poland; slawomir.dyjak@wat.edu.pl

**Keywords:** carbon gel, nitrogen-doped carbon materials, sol–gel synthesis, heteroatom doping, porous organic polymers, ultramicropores

## Abstract

Nitrogen-doped and heteroatom multi-doped carbon materials are considered excellent metal-free catalysts, superior catalyst supports for transition metal particles and single metal atoms (single-atom catalysts), as well as efficient sorbents for gas- and liquid-phase substances. Acid-catalyzed sol–gel polycondensation of hydroxybenzenes with heterocyclic aldehydes yields cross-linked thermosetting resins in the form of porous organic polymers (i.e., organic gels). Depending on the utilized hydroxybenzene (e.g., phenol, resorcinol, phloroglucinol, etc.) and heterocyclic aldehyde variety of heteroatom-doped organic polymers can be produced. Upon pyrolysis, highly porous and heteroatom-doped carbons are obtained. Herein, polycondensation of phloroglucinol with imidazole-2-carboxaldehyde (and other, similar heterocyclic aldehydes with two heteroatoms in the aromatic ring) is utilized to obtain porous, N-doped organic and carbon gels with N-content of up to 16.5 and 12 wt.%, respectively. Utilization of a heterocyclic aldehyde with two different heteroatoms yields dually-doped carbon materials. Upon pyrolysis, the porous polymers yield ultramicroporous N-doped and N,S co-doped carbons with specific surface areas of up to 800 m^2^g^−1^. The influence of the initial composition of reactants and the pyrolysis temperature on the structure and chemical composition of the final doped organic and carbon materials is studied in detail.

## 1. Introduction

Heteroatom-doped carbon materials are considered more chemically active than their heteroatom-free counterparts. Surface functionalization and substitutional doping of graphitic and graphenic carbon structures modify their intrinsic properties, including catalytic activity, wettability, and acid-base properties. Nitrogen doping is especially common because it yields carbons with intrinsic metal-free catalytic properties and also superb supports for carbon-supported transition metal catalysts, including metal nanoparticles, clusters, and single metal atoms/ions [1,2,3,4]. Indeed, N-doped carbons prove to be peerless supports for so-called single atom catalysts [5,6,7]. N-doped porous carbons are also considered efficient media for the separation, storage, and capture of gaseous substances such as CH_4_, CO_2_, or H_2_ [8,9,10,11]. There are many methods to obtain porous carbons with heteroatom doping, including the utilization of biomass, ammonia treatment, ionic liquids, or synthetic polymeric precursors. Synthetic precursors such as metal organic frameworks allow better control of porosity and doping in comparison to natural feedstock [12]. Unlike any other precursors, synthetic polymers allow the production of heteroatom-doped carbons of very specific characteristics, including a porous structure, morphology, and foreign atom doping type and position [13,14]. The range of possible synthetic polymers is large. We have shown that sol–gel polycondensation of resorcinol and various heterocyclic aldehydes such as 2-pyrrolaldehyde, 2-thiophenaldehyde or 2-formylselenophene yields thermosetting organic gels doped and co-doped with N, S or Se heteroatoms [15,16]. Upon pyrolysis, a whole range of heteroatom-doped porous carbons can be obtained from such organic polymers, with high yield. 

Zeolitic imidazolate framework MOFs (metal-organic frameworks, e.g., ZIF-8) became precursors of choice for the synthesis of microporous N-doped carbons [17], but easier and cheaper alternatives are still under scrutiny. Porous carbons in the form of gels (aero-, xero-, and cryogels) possess unique three-dimensional networked structures and hence constitute attractive candidates for catalysts, catalyst supports, and sorbents [18]. Herein, we propose alternative materials for the synthesis of N-doped carbons, based on imidazole-containing cross-linked porous organic polymers produced in the form of monolithic gels. Imidazole-based reagents are often utilized to produce porous N-doped carbons with very high doping contents [10,19,20]. Such materials have been studied inter alia, as (electro)catalysts for oxygen reduction reaction, and as electrodes for supercapacitors and lithium-ion batteries. Imidazole-based, cross-linked polymers have already been proposed as a source of porous carbons [8,9]. For instance, in a very unconventional approach, zeolitic imidazolate framework particles were covered by phenolic resin to produce a ZIF/resin nanocomposite. This unique composite was then transformed into nitrogen-coordinated cobalt-doped mesoporous carbon (Co−N−C), which showed high hydrogen evolution catalytic activity [21]. Indeed, N-doped porous carbons are able to coordinate single transition metal ions (e.g., Fe, Co, Cu, Mn, etc.) by pyridinic or pyrrolic nitrogen functionalities and yield unique catalytic materials of many advanced applications, including (electro)catalysis (as an alternative to expensive precious metals [22,23]), medicine (as artificial enzymes [24]), and separation and purification processes (e.g., as materials for wastewater remediation [25]). In this report, we show that polycondensation of phloroglucinol with imidazole-2-carboxaldehyde yields porous organic polymers and ultramicroporous carbons with N-content of up to 16 and 12 wt.%, respectively, while polycondensation with 2-thiazolecarboxaldehyde is a facile approach to synthesize N and S co-doped carbons (N,S-co-doping).

## 2. Materials and Methods

### 2.1. Materials

Imidazole-2-carboxaldehyde (>98.0%), 1-methylimidazole-2-carboxaldehyde (>98.0%) and 2-thiazolecarboxaldehyde (>97.0%) were obtained from the Tokyo Chemical Industry Co., Ltd. Resorcinol and phloroglucinol (>99.0%) were purchased from Sigma-Aldrich. The utilized solvents and acid (methyl alcohol (>99.8%), *N*,*N*-dimethylformamide (DMF; >98.0%), 2-propanol (>99.7%), hydrochloric acid (35–38%)) were obtained from Avantor Performance Materials Poland S.A. 

### 2.2. Synthesis

In order to obtain a hydroxybenzene–heterocyclic aldehyde organic gel, an acid-catalyzed (HCl_aq_) sol–gel polycondensation was conducted, and the molar ratio of hydroxybenzene/heterocyclic aldehyde was kept at 0.5 or 0.33. A schematic representation of the syntheses performed is presented in Figure 1. Polycondensation of phloroglucinol with imidazole-2-carboxaldehyde or 1-methylimidazole-2-carboxaldehyde yielded N-doped organic and carbon gels. To obtain dually-doped organic and carbon gels, a heterocyclic aldehyde with two different heteroatoms in the heterocyclic ring can be utilized. Accordingly, to obtain a N,S co-doped gel, 2-thiazolecarboxaldehyde was polycondensated with phloroglucinol. In a typical synthesis, 0.80 g of phloroglucinol and a corresponding amount of heterocyclic aldehyde were used. Organic ingredients were simultaneously dissolved in the solvent at room temperature using magnetic stirring and the sol–gel polycondensation was initiated by adding 1.0 mL of concentrated hydrochloric acid. For the mixture with imidazole-2-carboxaldehyde, the solution was heated to 50 °C to speed up aldehyde dissolution. The liquid mixture obtained was closed in a tight container and placed in an oven set at 55 °C. In the case of imidazole-2-carboxaldehyde, the sol to gel transition was almost instantaneous, while in the case of other aldehydes it took approximately 10–15 min. (please see Appendix A for details). Once obtained, the wet gels were aged for 24 h at 55 °C (in the tight container) and then dried at atmospheric pressure at 70 °C to a constant mass. Gels obtained with *N*,*N*-dimethylformamide (used as a solvent) were immersed in a large amount of acetone for 24 h prior to drying to remove the non-volatile DMF. Eventually, dark (or reddish) monoliths were obtained (Figure 2 and Figure 3). The dry organic gels were black in color, except the gel obtained from 2-thiazolecarboxaldehyde, which was light brown (Figure 3). The dry organic gels (xerogels) were subsequently carbonized at 750, 850 or 950 °C for 1 h under a stream of N_2_ at a heating rate of 5 °C min^−1^. The samples obtained from imidazole-2-carboxaldehyde were labeled CN−750, −850, or −950; those obtained from 1-methylimidazole-2-carboxaldehyde as CN(M)−750, −850, or −950; and those from 2-thiazolecarboxaldehyde as CNS−750, −850, or −950. Due to the highly cross-linked, thermosetting nature of the organic polymers, the pyrolysis yields were quite high, but varied in a broad range of 27−40%, depending on the heterocyclic ring and pyrolysis temperature (see Appendix A for details).

### 2.3. Characterization

N_2_ adsorption–desorption measurements were performed at liquid nitrogen temperature using an Autosorb IQ analyzer (Quantachrome Instruments, Florida, FL, USA). The samples were outgassed at 300 °C for 24 h prior to analysis. The Brunauer, Emmett and Teller specific surface area (S_BET_) was calculated from the N_2_ adsorption isotherm. The total pore volume (V_t_) was estimated from the volume adsorbed at a relative pressure (p/p_0_) of ~0.99, and the pore size distributions (PSDs) were calculated using the density functional theory for the carbon heterogeneous surfaces model (2D-NLDFT) embedded in the SAIEUS software. An elemental analysis to assess C, H, N and S wt.% content was performed using a Vario EL Cube apparatus (Elementar). A scanning electron microscope (SEM) operating at an accelerating voltage of 2kV (Carl Zeiss Ultra Plus) was used to study the carbon gels’ morphology. XRD analysis was performed on a Bruker D2 PHASER diffractometer with Cu Kα X-rays operating at 30 V and 10 mA. X-ray photoelectron spectroscopy (XPS) analysis was carried out using a PHI 5000 VersaProbe (Scanning ESCA Microprobe ULVAC-PHI) spectrometer with monochromatic Al Kα radiation (hν = 1486.6 eV) from an X-ray source (23 W). Both survey and the high-resolution XPS spectra were collected with the hemispherical analyzer at pass energies of 117.4 and 23.5 eV. The binding energy (BE) scale was referenced to the carbon C 1s peak with BE = 284.6 eV. Shirley background subtraction and peak fitting with Gaussian−Lorentzian-shaped profiles was performed for the high resolution XPS spectra using the CasaXPS software (version 2.3.16). XPS quantification of C, O, S, and N atomic concentrations was performed using standard sensitivity factors contained in the ULVAC-PHI MultiPak software (version 9.5.0). 

## 3. Results and Discussion

As shown in our previous reports, sol–gel polycondensation of hydroxybenzenes (phenol, resorcinol, etc.) and heterocyclic aldehydes yields a variety of heteroatom-doped porous organic polymers [15,16]. By varying the sol–gel transition conditions, the initial ratio of reactants (hydroxybenzene/aldehyde), and pyrolysis temperature, carbonaceous materials with tailored heteroatom-doping, structure and porosity can easily be obtained. As shown elsewhere, heterocyclic aldehydes with one heteroatom in the aromatic ring can react easily with resorcinol to yield porous organic polymers [16]. However, in the case of aldehydes based on five-membered heterocycles with two heteroatoms, the acid-catalyzed polycondensation with resorcinol to cross-linked polymers does not occur. As a less active reagent, resorcinol did not condensate with any of the three heterocyclic aldehydes studied herein, and for this reason resorcinol was replaced by much more reactive phloroglucinol. Heterocyclic aldehydes exhibit drastically different melting points and solubility depending on whether they can form hydrogen bonds. Imidazole-2-carboxaldehyde forms strong hydrogen bonds and hence exhibits a high melting point, low solubility, and volatility. For this reason, sol–gel polycondensation was performed in DMF—a highly polar solvent. In contrast, 1-methylimidazole-2-carboxaldehyde exhibits much better solubility, and in this case polycondensation was performed in methyl alcohol. Liquid 2-thiazolecarboxaldehyde is volatile and easily soluble, and hence the sol–gel synthesis was also performed in methanol. However, extensive optimization of the synthesis conditions showed that utilization of a single solvent is insufficient to obtain organic xerogels with well-preserve porosity. During drying under atmospheric pressure, the porosity of organic gels can totally collapse, creating non-porous solids, or it is retained, yielding porous solids (Figure 2). This depends on the presence of large mesopores and macropores in the original organic gel which allows drying without much shrinkage (shrinkage occurs due to capillary pressure). To prevent porosity collapse during drying, macroporosity can be induced in the organic gels by adding antisolvent to the sol–gel primary solvent to accelerate solvent−polymer phase separation. Consequently, isopropanol was used and added to the primary solvent as the antisolvent. DMF or methanol were mixed with isopropanol with a volume ratio of 2.5 (see Appendix A for details). 

SEM analysis (Figure 4) revealed the colloidal and macroporous morphology of the carbonaceous materials obtained with the addition of an antisolvent (isopropanol). While the macroporous polymers are very brittle and matt solids (Figure 2, on the right), the polymers obtained without antisolvent are totally non-porous, hard, and shiny compact solids (Figure 2, on the left). All the porous materials, regardless of the aldehyde utilized, exhibited a similar morphology of ~1 µm spherical particles infused into a three-dimensional macroporous architecture.

Table 1 presents the characteristics of the organic and carbon gels obtained via polycondensation (and then pyrolysis) of phloroglucinol and corresponding aldehydes (imidazole-2-carboxaldehyde (CN), 1-methylimidazole-2-carboxaldehyde (CN(M) or 2-thiazolecarboxaldehyde (CNS)) with a molar ratio of 1:3. Appendix A (Appendix A) shows the same parameters for materials obtained from mixtures with a phloroglucinol:aldehyde molar ratio of 1:2. The CN organic gel contained as much as ~16.6 wt.% of N, and this value decreased to 6.5 wt.% upon pyrolysis at 950 °C. The CN(M) organic gel contained a slightly smaller amount of N, however, after pyrolysis at 950 °C, 5 wt.% was still retained in the carbonaceous structure. The nitrogen and sulfur co-doped porous organic polymer contained ca 7.5 wt.% of N, and as much as 16.5 wt.% of S. Upon harsh carbonization at 950 °C, the heteroatom content decreased to 3 and 4 wt.%, respectively. Thermosetting resins constitute highly cross-linked polymers, which yield the so-called hard carbons (non-graphitizing carbons). The carbonaceous materials obtained via pyrolysis at temperature as high as 950 °C were still poorly graphitized, showing only two very broad reflections of low intensity in the XRD patterns (Figure 5). As non-graphitizing carbons, high-temperature (>2000 °C) graphitization of such materials would eventually yield the so-called glass-like carbons instead of graphite.

XPS analysis of the organic and corresponding carbonaceous materials revealed the presence and chemical state of the doped heteroatoms (Figure 5 shows a comparison of the XPS survey spectra of the organic and pyrolyzed N,S co-doped materials). High resolution XPS spectra of N 1s and S 2p of the co-doped organic and corresponding carbon gels obtained from 2-thiazolecarboxaldehyde via pyrolysis at 750 °C are presented in Figure 6. The spectra of nitrogen and sulfur elements in the organic polymers are quite simple, because they correspond to the thiazole heterocyclic ring (aromatic, contiguous −C−N−C− and −C−S−C− moieties). However, the chemical states of N and S became much more complex upon pyrolysis. It must be stressed here that XPS analysis, spectra deconvolution, and allocations of deconvoluted bands are prone to overinterpretations, misconceptions, and hence must be always read with caution [26,27,28]. This issue is especially relevant for porous, disordered carbonaceous materials, where heteroatoms occur in a variety of different chemical states. There is a large range of possible structures that can be assigned to C−C(H), C−N, C−S, or C−O bonding depending on the methodology/model applied. Upon pyrolysis, new forms of N-moieties appear in the N 1s spectrum (pyridinic N (398.2 eV), −NH_2_/pyridone (399.7 eV), pyrrolic N/or graphitic N (400.8 eV), and pyridine-N-oxide (402.7 eV) [26,27,28,29]. Sulfur occurs in the carbonaceous matrix, mainly as sulfide moieties (e.g., thiophenic −C−S−C− moieties) with a small contribution from oxidized sulfur states. 

The organic polymer derived from imidazole-2-carboxaldehyde exhibits a relatively simple N 1s XPS spectrum with the main peak around 401 eV, which can be assigned to the imidazole ring’s protonated pyrrolic nitrogen (Figure 7). Upon pyrolysis, numerous new forms of N-moieties (i.e., pyridinic, pyrrolic, graphitic, and pyridine-N-oxide) appear in the spectrum, just like in the case of N and S co-doped carbons discussed above [26,27,28,29]. High resolution C 1s XPS spectra of the organic and carbon gels (and the samples’ elemental surface compositions) can be found in the Appendix A (Appendix A). 

Figure 8 and Appendix A depict the low-temperature N_2_ adsorption–desorption isotherms of the carbon gels produced. An attempt to measure N_2_ adsorption of the solely N-doped carbons (CN and CN(M) samples) in the whole range of relative pressure (p/p_0_ of 10^−7^–1.0) failed (for reasons elucidated below). It was only possible to conduct this measurement for the N and S co-doped samples (CNS). Indeed, CN and CN(M) carbon xerogels doped only with N possess atypical N_2_ isotherms with wide and open low-pressure hystereses. Such hystereses indicate that the N_2_ physisorption measurements at −196 °C exhibit kinetic restrictions and cannot achieve equilibrium in any acceptable measurement time [30]. This phenomenon is caused by the presence of very narrow micropores. Carbons with small ultramicropores (<0.7 nm) cannot be characterized by the low-temperature N_2_ adsorption technique due to the restrictions for N_2_ diffusion into the narrowest micropores (kinetic restrictions at −196 °C). For these reasons, the low-temperature N_2_ isotherms for CN and CN(M) samples were collected only in the range of higher p/p_0_ values of ~10^−4^. The N-doped carbons (obtained from phloroglucinol:aldehyde mixtures with 1:3 molar ratio) exhibited relatively low S_BET_ values in the range of ca. 500–520 m^2^g^−1^ (Table 1). Moreover, the values were weakly affected by pyrolysis temperature. 

The situation is very different for carbons co-doped with sulfur. In this case, the isotherms were collected in the whole available p/p_0_ range. These carbons exhibited much higher S_BET_ values and the values grew significantly with the pyrolysis temperature—from 696 for CNS–750 to 810 m^2^g^−1^ for CNS–950. Due to the thermal stability of the –C–S–C– aromatic moiety, the porogenic properties of thiophenic sulfur can be observed at high carbonization temperatures, where the stable –C–S–C– aromatic moiety decomposes, and sulfur is lost as gaseous CS_2_, yielding the extensive microporosity of S-doped carbons. The PSD profiles (Figure 8) show that the microporosity of the N and S co-doped carbons (CNS samples) originated mainly from micropores of ca. 0.5−0.6 nm in size. The structure and size of the micropores did not change much in the 750−950 °C temperature range. These PSD profiles qualify the studied carbons as ultramicroporous solids. 

## 4. Conclusions

Heteroatom-doped carbonaceous materials are at the frontier of metal-free and single-atom catalysts [3,4]. Nitrogen-doped and co-doped carbons are of special interest, because N-doping allows the exceptionally stable coordination of transition metals on carbon supports [6]. For this reason, new forms of N-doped porous carbons are actively sought. Carbon gels remain at the center of researchers’ attention due to their unique porous structure and ease of preparation [31,32]. Here, we proposed another versatile approach to synthesize microporous carbons with a high number of heteroatoms, which can easily be obtained from porous organic polymers. Polymers with N and S content of up to 16.5 wt.% can be turned into microporous carbons with a yield of up to 40% and specific surface areas of up to 800 m^2^g^−1^. Importantly, a comparison of the characteristics of organic and carbonaceous materials obtained from mixtures with a phloroglucinol:aldehyde molar ratio of 1:2 and 1:3 revealed no significant differences, and hence a phloroglucinol:aldehyde molar ratio of 1:2 or 1:2.5 can be considered as optimal (a higher content of heterocyclic aldehydes is already excessive from the heteroatom-content point of view). A molar ratio lower than 1:2 would, on the other hand, result in poor cross-linking. Resorcinol was found to be less reactive than phloroglucinol, and hence not appropriate for polycondensation with heterocyclic aldehydes with two heteroatoms in the aromatic ring. The solely N-doped carbon gels exhibited very specific, atypical microporosity of narrow ultramicropores, and hence relatively low S_BET_ values (as determined by N_2_ physisorption at liquid nitrogen temperature), independent of the pyrolysis temperature. In contrast, N,S co-doped carbon gels possessed higher specific surface area values, which grew significantly with increasing carbonization temperatures. Macroporosity can be induced in the organic and carbon gels by adding antisolvent (isopropanol in this case) to the primary mixture of reagents. The high content of nitrogen and unique porous structure of this new kind of carbon-based materials make them promising candidates for applications related to gas storage/capture and energy storage/conversion. For instance, such materials may be an interesting alternative for the commonly utilized zeolitic imidazolate framework eight (ZIF-8), and due to their high N-content they may be probed as a new type of metal-free catalyst or as efficient supports for transition metals. This is especially important, because transition metals supported on N-doped carbons (the so-called TM−N−C materials) have attracted enormous attention as platinum group metal-free catalysts, artificial enzymes, and gas storage media—to name a few prospective, advanced applications. 

## Figures and Tables

**Figure 1 molecules-26-00668-f001:**
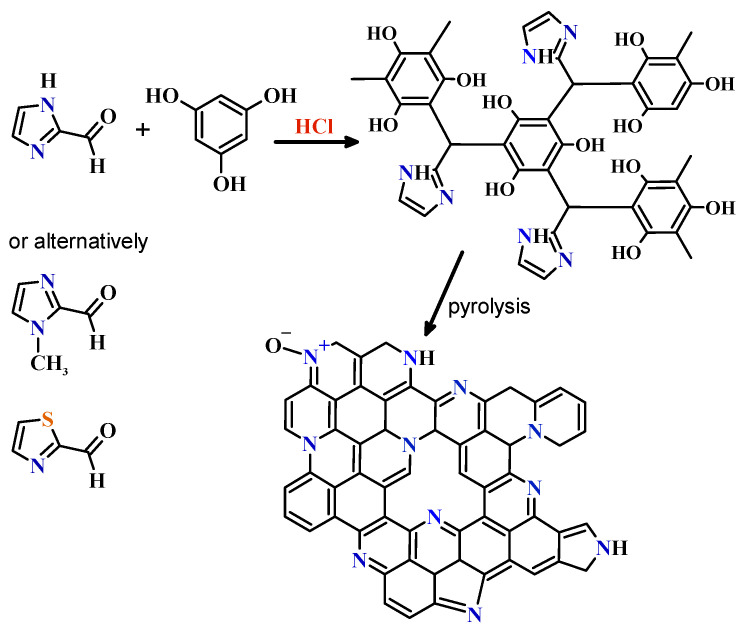
Schematic representation of imidazole-based N-doped carbons synthesis (1-methylimidazole-2-carboxaldehyde and 2-thiazolecarboxaldehyde can be used instead of imidazole-2-carboxaldehyde).

**Figure 2 molecules-26-00668-f002:**
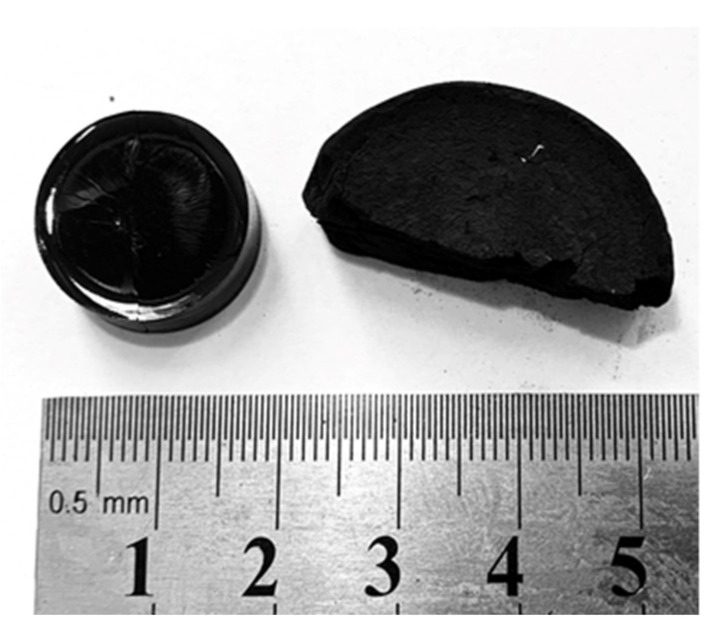
(**Left**) A non-porous dry organic xerogel upon collapse of nanoporosity (significant drying shrinkage) and (**right**) corresponding macroporous xerogel (obtained with the addition of isopropanol as an antisolvent), which did not undergo collapse of porosity and shrinkage during drying.

**Figure 3 molecules-26-00668-f003:**
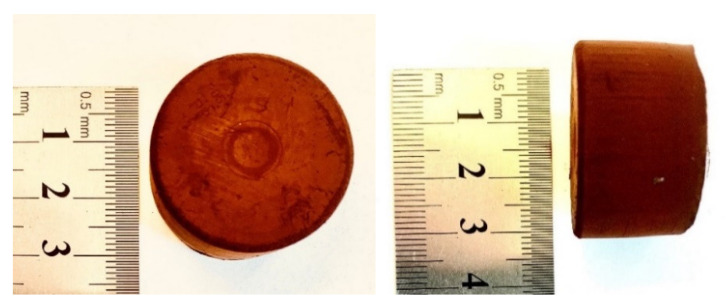
The dry 2-thiazolecarboxaldehyde-based monolithic organic gel (organic xerogel).

**Figure 4 molecules-26-00668-f004:**
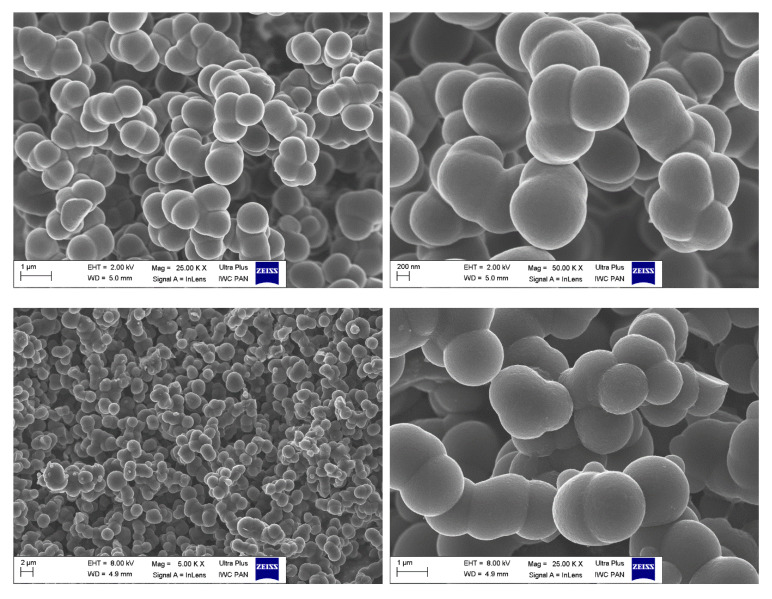
SEM micrographs showing the typical morphology of the sol–gel derived N (and S) co-doped carbon gels—sample CNS–950 (top row) and sample CN–950 (bottom row).

**Figure 5 molecules-26-00668-f005:**
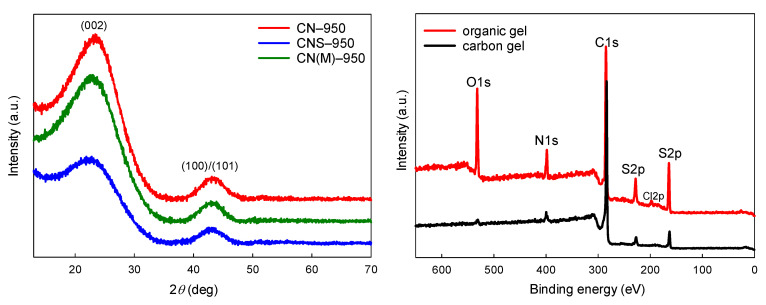
XRD patterns of carbonaceous materials obtained from the porous polymers after pyrolysis at 950 °C. XPS survey spectra of organic and corresponding carbon gels (pyrolysis at 750 °C) obtained from 2-thiazolecarboxaldehyde.

**Figure 6 molecules-26-00668-f006:**
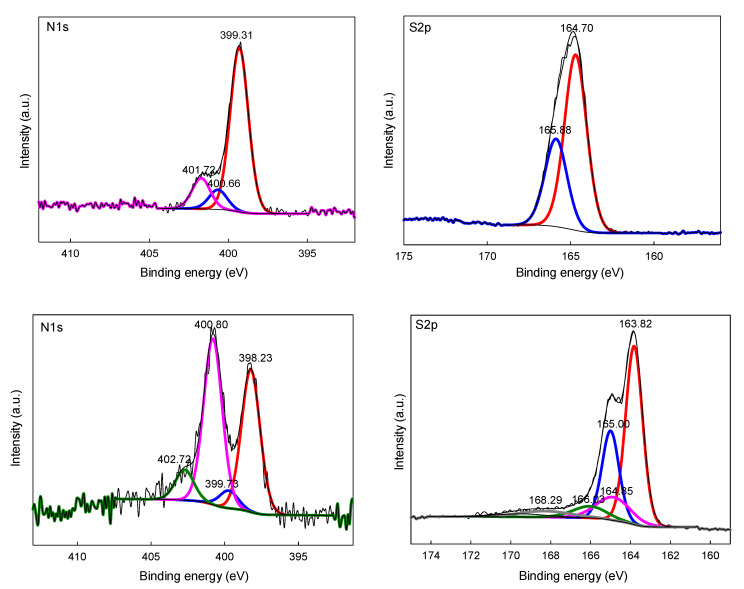
High resolution XPS spectra of N 1s and S 2p of organic (top row) and corresponding carbon (bottom row) gels obtained from 2-thiazolecarboxaldehyde via pyrolysis at 750 °C.

**Figure 7 molecules-26-00668-f007:**
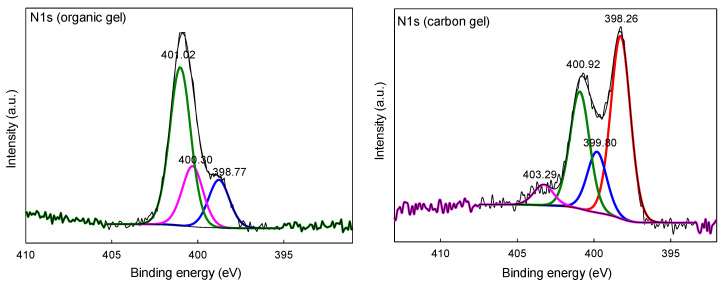
Comparison of high resolution XPS spectra of N 1s of organic and corresponding carbon gels obtained from imidazole-2-carboxaldehyde via pyrolysis at 750 °C.

**Figure 8 molecules-26-00668-f008:**
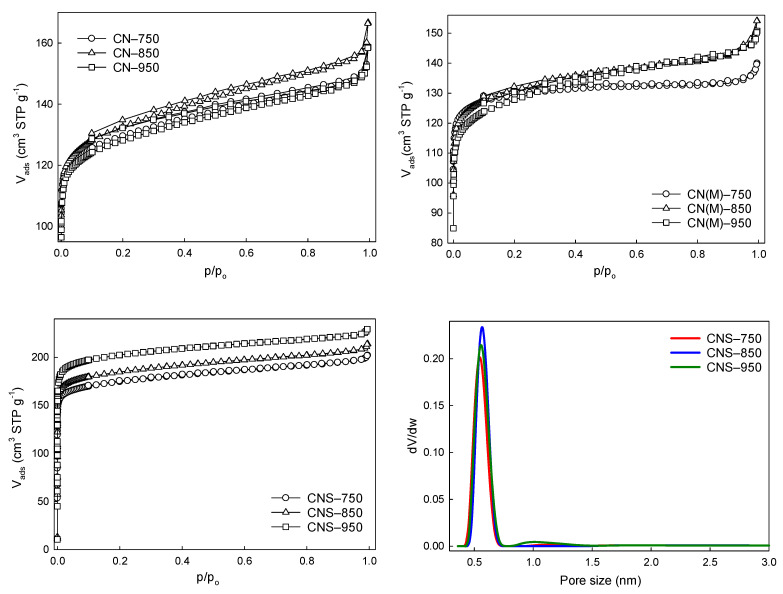
N_2_ adsorption–desorption isotherms of CN, CN(M) and CNS carbon gels obtained from a phloroglucinol:aldehyde molar ratio of 1:3 and pyrolyzed at three different temperatures. Pore size distribution (PSD) plots are also presented for the CNS carbon samples.

**Table 1 molecules-26-00668-t001:** Elemental analysis and textural characteristics of the organic and carbon gels obtained from a phloroglucinol:aldehyde molar ratio of 1:3.

Sample	N (wt.%)	C (wt.%)	H (wt.%)	S (wt.%)	S_BET_ (m^2^ g^−1^)	V_t_ (cm^3^ g^−1^)
**CN**	16.56	44.73	4.61	-	-	-
**CN–750**	12.01	71.50	1.96	-	511	0.25
**CN–850**	9.38	76.42	1.60	-	519	0.26
**CN–950**	6.47	81.20	1.27	-	505	0.24
**CN(M)**	13.93	44.02	5.38	-	-	-
**CN(M)–750**	9.43	74.01	2.06	-	524	0.22
**CN(M)–850**	7.24	75.57	1.90	-	520	0.23
**CN(M)–950**	4.91	77.70	1.79	-	501	0.23
**CNS**	7.40	41.72	3.55	16.44	-	-
**CNS–750**	5.18	69.65	2.10	7.22	696	0.31
**CNS–850**	4.17	71.31	2.06	5.50	735	0.33
**CNS–950**	3.11	72.77	2.05	4.02	810	0.35

## Data Availability

Data are available from the authors on request.

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
