# Peer review of "Nitrogen-Doped Carbons Derived from Imidazole-Based Cross-Linked Porous Organic Polymers"

_molecules, 2021, doi:10.3390/molecules26030668_

Round 1

Reviewer 1 Report

In this work, highly porous heteroatom doped carbon materials were synthesized via the pyrolysis of organic polymers. After pyrolysis, N-doped (or N, S-co-doped) porous carbons with high heteroatom-contents were obtained. Different heteroatoms can be induced into the carbons by tuning the species of heterocyclic aldehydes. The manuscript is well written and organized. Therefore, I recommend this paper could be accepted after minor revision.

  1. In this paper, heteroatom-doped carbons were obtained through polycondensation and pyrolysis methods. Could we directly anneal the materials containing C, N, and S elements, such as melamine and cysteine, to obtain the heteroatom-doped carbons?
  2. In Figure 8, three-dimensional pore size distribution plots are difficult to compare the pore size among the three simples. Two-dimensional plots are better.

Author Response

We would like to express our gratitude to all the reviewers for the time and effort they invested to improve our manuscript; we highly appreciate all the given suggestions.

  1. In this paper, heteroatom-doped carbons were obtained through polycondensation and pyrolysis methods. Could we directly anneal the materials containing C, N, and S elements, such as melamine and cysteine, to obtain the heteroatom-doped carbons?

RESPONSE: Yes it is interesting question to consider. In our manuscript we highlighted the fact that the obtained organic polymers constitute highly cross-linked, thermosetting resins and for this very reason they can be pyrolyzed (carbonized) and turned into carbonaceous materials with reasonably high yield. Similarly, biomass (e.g., wood made of, inter alia, lignin) is usually carbonizable with high yield as it constitutes some king of cross-linked polymeric materials. Lignin is in reality cross-linked phenolic resin – a bit similar to the polymers we presented in this manuscript. On the other hand, substances such as  melamine and cysteine (or urea for instance) can either easily evaporate or decompose into gases and hence carbonization may not be possible. Melamine is especially interesting case as during high-temperature treatment it condensates into polymeric substance called melon (via melem) but eventually upon heating above around 800 oC it would decompose into gaseous products. Ionic liquids could be a good example of non-polymeric substances, which can be carbonized with high yield due to their very low volatility and high thermostability.

  1. In Figure 8, three-dimensional pore size distribution plots are difficult to compare the pore size among the three simples. Two-dimensional plots are better.

RESPONSE: Yes, two-dimensional plots are indeed much better for comparison. We have made the necessary changes.

Reviewer 2 Report

In this manuscript, the porous carbons were obtained by pyrolysis of the products derived from polycondensation of phloroglucinol with different heterocyclic aldehydes containing N or N/S atoms. The influence of the initial composition of reactants and the pyrolysis temperature on the structure and chemical composition for the final carbon materials was also investigated. After minor revision, the paper can be published.

  1. In the “1. Materials and synthesis” part, the authors should write the content in different paragraphs to make the synthesis process to be more clear for the reader. Also, the discussion of the reasons why you choose the chemicals, reaction conditions, …, can be moved to the “Results and discussion” part.
  2. Can the authors provide some practical applications for these porous carbons in the catalysis or separation field to indicate their advantages?
  3. For XRD, we usually call it “XRD patterns”, not “XRD spectra”.

Author Response

We would like to express our gratitude to all the reviewers for the time and effort they invested to improve our manuscript; we highly appreciate all the given suggestions.

  1. In the “1. Materials and synthesis” part, the authors should write the content in different paragraphs to make the synthesis process to be more clear for the reader. Also, the discussion of the reasons why you choose the chemicals, reaction conditions, …, can be moved to the “Results and discussion” part.

RESPONSE: As suggested, we did make significant changes to the Materials and synthesis description and all the changes are labeled (highlighted) in the revised manuscript. Now the parts devoted to materials, syntheses and discussion are well separated.

  1. Can the authors provide some practical applications for these porous carbons in the catalysis or separation field to indicate their advantages?

RESPONSE: In this report we have focused our effort on synthesis (at various conditions) and characterization of the new carbon-based materials. To precisely scrutinize the synthetic conditions and their influence on the final properties of the new kind of (co)doped carbons we did not explore applications. However, due to very high N-content and developed porosity these materials are promising media for gas storage and they can also serve as supports for preparation of heterogeneous catalysts where metal nanoparticles and single atoms are supported on N-doped carbons. The higher the N-content the more active metallic centers can be created. Our previous research on resorcinol/2-pyrrolaldehyde gels has already shown that such materials are good candidates for the so-called platinum group metal-free (non-precious metal) catalysts to be used as a replacement of Pt in polymer electrolyte fuel cells (please see e.g., Enhancement of PGM-free oxygen reduction electrocatalyst performance for conventional and enzymatic fuel cells: The influence of an external magnetic field, https://doi.org/10.1016/j.apcatb.2019.117955 or Heterogeneous iron-containing carbon gels as catalysts for oxygen electroreduction: Multifunctional role of sulfur in the formation of efficient systems, https://doi.org/10.1016/j.carbon.2017.02.045). These new carbons obtained from imidazole-based aldehydes presented in this manuscript contain even more nitrogen that the materials presented in the above-mentioned papers, and as such they can perform even better as PGM-free electrocatalyst upon doping with iron. Additionally, such carbons, upon doping with transition metals can serve as the so-called artificial enzymes (please see for instance https://doi.org/10.3390/ma13173707). We will explore this research directions in near future, however such research will demand extensive laboratory work, and hence maybe it is better to present it in separate report. Also, the very narrow microporosity of the N-doped carbons may be promising for H2 storage and CO2 capture as we hinted in our previous attempts (https://doi.org/10.1016/j.micromeso.2015.11.050). However, herein in this report we wanted to focus our attention especially on the synthesis-properties correlations. We added some new references and discussions in the Introduction and Conclusions to show the perspective applications of this presented materials.

  1. For XRD, we usually call it “XRD patterns”, not “XRD spectra”.

RESPONSE: The description was changed from spectra to patterns

Reviewer 3 Report

The manuscript reports on the preparation of N- or S-doped carbon by pyrolysis of gel formed by condensation of phloroglucinol with imidazole-2-carboxaldehyde. The data fully support the conclusions and the topic is surely interesting and on the cutting edge of research in the field. For these reason the manuscript can be accepted for publication almost “as is”. The unique suggestion is to better highlight the novelty of the approach (in the introduction) and of the obtained results (in the conclusions) with respect to previous work already cited by the authors and adding the following:

Sang et al. New J. Chem., 2019,43, 3078-3083

Where an imidazole derivative was used with phloroglucinol to obtain N-doped mesoporous carbons

Author Response

We would like to express our gratitude to all the reviewers for the time and effort they invested to improve our manuscript; we highly appreciate all the given suggestions.

  1. The unique suggestion is to better highlight the novelty of the approach (in the introduction) and of the obtained results (in the conclusions) with respect to previous work already cited by the authors and adding the following: Sang et al. New J. Chem., 2019,43, 3078-3083. Where an imidazole derivative was used with phloroglucinol to obtain N-doped mesoporous carbons

RESPONSE: Indeed, the paper by X. Sang, J. Chen, M. Jing, G. Shi, C. Ni, D. Wanga, Wei Jin, Sustainable synthesis of nitrogen-doped porous carbon with improved electrocatalytic performance for hydrogen evolution. New J. Chem., 2019, 43, 3078-3083, https://doi.org/10.1039/C8NJ05819A, fits perfectly the list of our references and as such it was added to the manuscript (in the Introduction part). We also slightly modified the Introduction/Conclusions to highlight the novelty of the presented research and its potential in future, possible real-world applications.